# Respiratory Viruses in Nosocomial Pneumonia: An Evolving Paradigm

**DOI:** 10.3390/v15081676

**Published:** 2023-07-31

**Authors:** Marya D. Zilberbeg, Imran Khan, Andrew F. Shorr

**Affiliations:** 1EviMed Research Group, LLC, P.O. Box 303, Goshen, MA 01032, USA; evimedgroup@gmail.com; 2Pulmonary and Critical Care Medicine, Medstar Washington Hospital Center, Washington, DC 20010, USA; imran.khan@medstar.net

**Keywords:** bacteria, hospital, nosocomial, pneumonia, virus

## Abstract

Nosocomial pneumonia (NP) represents a leading cause of morbidity and mortality in hospitalized patients. Historically, clinicians have considered hospital-acquired pneumonia (HAP) and ventilator-associated pneumonia (VAP), which comprise NP, to be essentially bacterial processes. As such, patients suspected of having either HAP or VAP are initially treated with broad-spectrum antibiotics, and few clinicians search for a possible culprit virus. Recent reports which build on earlier studies, however, indicate that viruses likely play an important role in NP. Studies employing viral diagnostics as part of the evaluation for NP indicate that common respiratory viruses can spread nosocomially and lead to HAP and VAP. Similarly, studies of the general epidemiology of respiratory viral infections, such as influenza, respiratory syncytial virus, adenovirus, and rhinovirus, confirm that these pathogens are important causes of NP, especially among immunosuppressed and pediatric patients. More importantly, these more contemporary analyses reveal that one cannot, based on clinical characteristics, distinguish a viral from a bacterial cause of NP. Additionally, viral HAP and VAP result in crude mortality rates that rival or exceed those reported in bacterial NP. Rigorous prospective, multicenter trials are needed to confirm the significance of respiratory viruses in NP, as are studies of novel therapeutics for these viral infections.

## 1. Introduction

Hospital-acquired infections (HAIs) represent a major challenge, costing US hospitals nearly $30 billion annually [1]. Nosocomial pneumonia (NP), comprising both hospital-acquired pneumonia (HAP) and ventilator-associated pneumonia (VAP), remains the second most prevalent HAI and leads to substantial antimicrobial use [2]. HAP represents new pneumonia developing in a patient after more than 48 h of hospitalization, while VAP is a subtype of HAP that is defined as new pneumonia diagnosed after a patient has received mechanical ventilation (MV) for more than 48 h [2]. Presently, the incidence of HAP approaches five to 10 cases per 1000 admissions, while VAP occurs five to 10 times more frequently than HAP [2,3]. In contrast to some other HAIs, such as surgical site infections, HAP and VAP are associated with significant crude mortality and substantial morbidity [3]. Although controversy exists surrounding the attributable mortality of NP, these infections prolong both the duration of MV and a patient’s length of stay in the intensive care unit (ICU) and in the hospital. As a consequence, the costs of NP to the healthcare system are extreme [4]. As such, preventing HAP and VAP represents a major focus of hospital-based quality initiatives.

Traditionally, HAP and VAP have been viewed as bacterial infections. A range of bacteria, including many difficult-to-treat pathogens such as *Pseudomonas aeruginosa* (PA), *Acinetobacter baumannii* (AB), and methicillin-resistant *Staphylococcus aureus* (MRSA), can cause NP [2,3]. Not surprisingly, therefore, the timely administration of in vitro active antibiotics remains one of the most important determinants of outcomes in NP [5]. However, many patients with NP never have cultures that reveal evidence of a bacterial organism. In some cases, this arises because either there is no true infection or because the subject has suffered an aspiration event which has not resulted in bacterial superinfection.

Although respiratory viruses have long been recognized as causes of pneumonia and respiratory failure in the community setting, more recent evidence identifies them as etiologic agents in NP. For example, such viruses as influenza, respiratory syncytial virus (RSV), human metapneumovirus (hMPV), and rhinovirus have all been reported to trigger pneumonia syndromes clinically indistinguishable from other causes of HAP and VAP [6,7]. It is likely that at least some of this increase in the recognition of these viruses as causative pathogens in NP reflects improved viral diagnostic tools brought to the market in recent years. 

More significantly, the fact that respiratory viruses can spread nosocomially should not surprise any clinician who cared for patients suffering from COVID-19 [8,9]. During the pandemic, many studies documented that this virus could spread to hospitalized patients either from other patients or from asymptomatic family members and healthcare providers [8,9]. This led to some previously uninfected patients admitted for non-COVID-19 related illnesses dying from COVID-19-induced acute respiratory distress syndrome (ARDS).

In light of this enhanced appreciation for the potential nosocomial transmission of respiratory viruses coupled with better diagnostic techniques for these organisms, physicians caring for the critically ill require an understanding of the epidemiology and burden of viruses as causes of HAP and VAP. Greater attention to diagnosis and treatment of these pathogens may yield not only better individual patient outcomes but also lead to a decrease in antimicrobial overuse and curtail the escalation of antimicrobial resistance.

As a context for the present review, studies examining this topic generally take several forms. Some reports are designed syndromically, focusing on patients with HAP and VAP, and describe the results of various forms of viral testing in patients with these syndromes. Other analyses are pathogen-oriented, describing viral microbiology and presenting the prevalence of various organisms across a broad range of infections, of which community-acquired pneumonia (CAP) and NP are a subset. While investigators have conducted most of these epidemiology studies in adult patients, some reports focus specifically on the pediatric population. Furthermore, irrespective of the study design, it remains critical to understand viral NP in immunosuppressed patients. These individuals represent a distinct subgroup at higher risk for viral HAP and VAP.

## 2. Materials and Methods

We prepared a narrative review of recent studies on viral NP. We searched Medline from 2017 to 2023 with the keyword search terms: hospital-acquired pneumonia, nosocomial pneumonia, and ventilator-associated pneumonia, along with the terms: epidemiology, microbiology, and virus. We identified articles focusing specifically on NP, in addition to reports addressing specific individual viral pathogens. The studies included in our analysis all applied the standard definitions of HAP and VAP as described above.

## 3. HAP and VAP Microbiology Studies

Several reviews prior to the COVID-19 pandemic discussed the importance of respiratory viruses in NP. Since these earlier summaries of the literature, newer studies have bolstered the hypothesis that viruses can be important causes of HAP and VAP [6,7]. For example, Loubet and colleagues evaluated 143 patients with NP in a single ICU, of whom approximately two-thirds underwent testing with a multiplex polymerase chain reaction (mPCR) assay [10]. Among those evaluated, 32% tested positive for a viral infection. Approximately 57% of these cases were infected only with a virus. In the remaining 43%, there was a bacterial and viral coinfection. The most common viruses identified included influenza, rhinovirus, and RSV. The majority of viruses were seen in the immunocompromised [10]. Notably, there was little difference in mortality based on the type of infecting pathogen (e.g., virus vs bacteria). Although somewhat limited because of its single centre and retrospective design, this study importantly confirms the significance of viruses in HAP and VAP.

In a similar analysis focusing on patients with ARDS, Shorr et al. reported that viruses alone were isolated in 21.7% of individuals with acute respiratory failure [11]. The majority of these events represented CAP. However, a substantial number of patients with HAP and VAP who progressed to ARDS were noted to have an acute viral infection. Consistent with the observations of Loubet et al., viruses occurred more often in the immunosuppressed [10]. Importantly, crude hospital mortality associated with viral NP was substantial, in excess of 35% [11]. This mortality rate was similar to that seen in patients with bacterial NP. This suggests that bacterial and viral NP can result in similarly high mortality rates. Given the lack of a control population without NP in this analysis, one cannot assess the relative attributable mortality of viral and bacterial NP. While selection bias remains a major concern with this report (viral testing was not required in all cases), this investigation adds to our understanding of the significance of viruses in NP.

Examining HAP and VAP in aggregate may not be appropriate, as these syndromes likely have different pathophysiology. Additionally, patients diagnosed with VAP often undergo more extensive diagnostic testing, given that they are ventilated and cared for in an ICU. Addressing these infections separately, therefore, may provide more precise insights into the impact of respiratory viruses in the hospital. In one of the largest VAP cohorts evaluated (*n* = 710), viruses were only identified in 5.1% of the cases [12]. The most frequently recovered viruses included rhinovirus, RSV, adenovirus, and hMPV. Both immunosuppression and stem-cell transplantation were independently associated with a viral (as opposed to bacterial) etiology for VAP. These variables, though, performed poorly as screening tests for viral VAP [12]. This fact indicates that restricting testing for viruses to only those with immune system compromise would lead to underdiagnosing viral VAP. Although also a single-centre, retrospective analysis, this study is unique in that nearly 80% of those with VAP underwent mPCR viral testing. This limits a key form of bias and suggests that the lower prevalence of VAP in this study reflects a better approximation of the epidemiology of respiratory viruses in VAP. Readers should note that in addition to the bias introduced because not all patients underwent viral evaluation, bacterial testing methods also have their limitations. Bacterial cultures, even if taken from the lower airway, may prove imprecise.

Conversely, in a multicenter report dealing with non-ICU patients with NP, Jang and co-workers examined 379 subjects [13]. While a culprit organism was only identified in approximately one-third of the population, respiratory viruses accounted for 5% of all pathogens identified [13]. This number, although deriving from a non-ventilated population, is strikingly consistent with the VAP study described earlier. In contrast to prior reports, the parainfluenza virus represented the most commonly recovered virus. Moreover, distinct is the multicenter nature of this registry substantially enhances its generalizability and reveals more complex initiatives are possible for evaluating the epidemiology of viral NP [13].

In sum, multiple recent investigations confirm that respiratory viruses remain an important cause of NP, be it HAP or VAP. These analyses further establish that one cannot distinguish viral causes from bacterial etiologies based purely on clinical characteristics. Importantly, the mortality rate associated with viral NP remains high and, in part, likely reflects the lack of therapeutic options for patients with a viral respiratory infection.

## 4. Studies of Specific Viruses

As an alternative to describing the prevalence of respiratory viruses relative to other etiologies for HAP and VAP, one can explore this issue by focusing on the epidemiology of specific viral cases of pneumonia cared for in the hospital. In this way, it becomes possible to put the burden of a particular virus in perspective relative to its impact on community-dwelling subjects as opposed to persons who develop infection while hospitalized. Godoy et al., for instance, investigated influenza over several respiratory seasons in Spain so as to describe its impact on the healthcare system [14]. Over 6 years, these authors identified 1722 cases of influenza across 12 hospitals. In 96 subjects (5.1%), the infection was nosocomial in onset [14]. Influenza A accounted for nearly 85% of all events. In many instances of influenza NP, the disease began more than 14 days after admission. Patients with influenza NP (as opposed to influenza CAP) were older, had more chronic illnesses, and were more likely to be immunodeficient. Not surprisingly, persons with influenza NP faced a higher risk for death than those with community-onset infection (19 vs 13%, respectively [14]).

In a well-designed prospective observational analysis, Qalla—Widmer and co-workers confirmed the general findings of Godoy et al. [14,15]. Across over 5000 subjects hospitalized for influenza, these researchers identified 836 instances where influenza infection began more than 72 h after hospital admission [15]. This resulted in an incidence of 0.5 influenza NP cases per 1000 hospitalizations. It is important to note that in the two participating institutions, hospital workers were not required to undergo influenza vaccination. In addition, the authors knew little regarding the vaccination status of those who developed nosocomial influenza. These factors certainly limit the external validity of the findings and suggest they may not apply in instances where vaccine rates are higher. It is further important to stress the importance of healthcare worker vaccination as a tool against the nosocomial viral spread. For example, during the recent pandemic, vaccination against Severe Acute Respiratory Syndrome Coronavirus 2 (SARS-CoV-2) proved crucial in limiting the spread of this virus inside hospitals [16,17]. Unfortunately, no information was provided about the underlying immune status of any of the persons suffering from nosocomial influenza.

Similarly, Kim and colleagues prospectively evaluated 2865 patients with severe pneumonia in a Korean ICU over a decade, from 2010 to 2019 [18]. Slightly fewer than half of those enrolled were classified as NP. These researchers recovered influenza and RSV in 3.8% and 3.5% of severe NP cases, respectively [18]. Other common viral pathogens included hMPV, parainfluenza virus, and rhinovirus. Just as in earlier studies where clinical features did not differentiate persons with viral NP as opposed to bacterial NP, signs and symptoms of infection were similar between cases of severe influenza and severe RSV in ICU patients with NP. Mortality also did not differ based on the type of viral NP. The crude mortality in these syndromes approached 40% [18]. Put simply, although respiratory viruses predominantly result in CAP, multiple studies focused on both the general microbiology of HAP and VAP and the epidemiology of select, specific respiratory viruses illustrate that such respiratory viruses can result in NP. More importantly, instances of viral NP are clinically indistinguishable from those caused by bacterial pathogens, which suggests the need for broader use of viral diagnostics in the evaluation of suspected HAP and VAP – particularly in the immunosuppressed.

## 5. Pediatric Viral NP

The data exploring viral NP in pediatric populations are sparser than those in adults. Nonetheless, two reports highlight that viral NP is likely more prevalent and more of an issue among pediatric patients.

In an 8-month prospective study of over 4000 hospitalized pediatric patients in Mexico, Torres—Garcia et al. determined that only 2% developed NP, but fully 65.0% of events were attributed to a virus. This contrasts substantially with the rates observed in adults [19]. What might explain this difference? Immunosuppression does not appear to play a role, as in this study, the prevalence of immunosuppression was similar to what was reported in studies of adults and viral NP. In addition, children often lack many of the chronic co-morbid illnesses that affect adults, and which alter their risk for viral infection. However, children also lack the acquired immunity present in adults from prior respiratory viral infections — and this lack of acquired immunity may contribute to the higher prevalence of these pathogens in pediatric cases of NP. Alternatively, more than one-third of those examined by Torres—Garcia suffered from congenital heart disease or other congenital abnormalities [19]. That the authors performed viral testing on all patients with NP, however, represents the most likely explanation for the observed disparity. This fact additionally reveals a key strength of this analysis.

Despite the disparity in the prevalence of viral NP between children and adults, there appears to be little difference in the major respiratory viruses involved based on the findings of Torres—Garcia et al. Specifically, the key viruses in this pediatric cohort included RSV, influenza, and rhinovirus [19]. Although the mortality rate in this study was low, again likely reflecting the age of the patients, the need for MV was significant. Half of the viral NP population eventually required MV for pneumonia.

An additional study confirms the greater incidence of viruses in NP among children. Debbagh and co-workers conducted a small, single-centre analysis of pediatric VAP, which explored viral etiologies in 38 children [20]. Extensive testing of bronchial secretions with mPCR demonstrated the presence of a virus in the lower airway in 30.7% of patients [20]. In contrast to the report by Torres—Garcia, there were few immunosuppressed patients in this population. Furthermore, this study was conducted during the summer months, as opposed to during the traditional respiratory viral season, explaining the absence of any cases of influenza. The most common viruses isolated were rhinovirus and RSV. This pattern of viral recovery demonstrates an important point also described by both Kim et al. and by Shorr et al. in adults: NP due to influenza not surprisingly peaks during winter months while the prevalence of other viruses appears to display less seasonal variability [12,18,20].

## 6. Discussion

Why should clinicians care about the potential for certain viruses to cause HAP and VAP? On the one hand, the identification of a virus as the only potential pathogen in a patient clinically suffering from NP represents an opportunity for prudent antibiotic stewardship. Although this may be of limited significance if the prevalence of viruses in NP is low, the current explosion in antimicrobial resistance, particularly among Gram-negative bacteria, has been driven by inappropriately high rates of antibiotic prescription [21]. This, in turn, leads to a vicious cycle such that physicians employ ever broader antimicrobials. As a consequence, this results in escalating selection pressure favouring the emergence of extensively drug-resistant organisms. Diagnosing solitary viral organisms can allow the clinician to discontinue unneeded antibiotics. Put another way, concluding that a case of NP is attributable to a virus can foster timely antibiotic de-escalation. Prospective studies may be needed to examine the outcomes of patients with single-pathogen viral NP from whom antibacterial treatments are withheld before clinicians are comfortable with such practices.

Another potential advantage of viral testing is that for certain viruses, such as influenza, treatment options exist. Although limited in their efficacy, specifically in critically ill patients, inadvertently withholding a potential antiviral treatment could result in excess mortality and morbidity. This concern becomes even more pressing as multiple promising therapeutics are under study for RSV while novel agents are being developed to augment the armamentarium for treating influenza.

In the immunosuppressed, the issues raised above are that much more pressing. As immunocompromised patients face both an excess risk for viral NP and mortality from these infections, early identification of a virus can result in the prompter administration of appropriate supportive care along with the discontinuation of unnecessary antibiotics. As immunocompromised persons are often exposed to antibiotics and thus regularly become colonized with highly resistant bacteria, any effort to limit their treatment with antibiotics can help to prevent the dilemma where such a patient becomes infected with a bacterium for which we have no treatment alternatives [22]. Furthermore, as respiratory viruses can spread easily in a transplant ward or ICU, detection of a viral cause for HAP and VAP can prompt faster initiation of infection control measures to prevent mini outbreaks of these respiratory viruses. For example, viral infection necessitates droplet and the like precautions, while the presumption of a bacterial pathogen might trigger only contact precautions [23,24]. The identification of a culprit virus, therefore, will significantly alter infection control practices and, in turn, help stem the spread of disease and its attendant sequelae.

It is important to note that controversy remains regarding the clinical significance of isolating certain viruses from the respiratory tract in acutely ill patients. The debate about the importance of viruses in NP, though, generally relates to the reactivation of latent viruses such as the herpes simplex virus (HSV) [25]. HSV reactivation occurs commonly in critically ill patients. When isolated from the lower airways of patients, it remains uncertain if this represents true infection or just upper airway contamination [26]. Furthermore, it has been difficult to disentangle whether HSV, in such instances, simply represents a marker of more severe illness or if its recovery is accompanied by an independent increase in the risk for morbidity and mortality.

Reactivation for the respiratory viruses reviewed here (e.g., influenza, RSV, rhinovirus, etc.) is, though, not an issue. Multiple reports indicate that each of these viruses can cause severe illness in patients presenting to the hospital with CAP [27,28]. Moreover, in the articles reviewed above, the definitions of HAP and VAP necessitated that the patient meets standard clinical and radiographic criteria for new pneumonia. Hence, it seems unlikely that these viruses represent either only colonizers or episodes of reactivation. In other words, nearly all of Koch’s postulates have been satisfied for these respiratory viruses relative to their role in NP [29]. One caveat, though, bears mention. It remains possible that in some cases classified as viral NP, the patient could have had a subclinical infection at initial hospital presentation that only became truly evident several days into hospitalization. However, given that in many of the cases of viral NP described in the reviewed articles, symptoms only began after more than three days into a hospital stay, the potential for this seems to be less of a concern.

## 7. Conclusions

In summary, clinicians need to be aware that respiratory viruses often cause both HAP and VAP. Viral NP is associated with both substantial morbidity and mortality and is clinically indistinguishable from bacterial cases of NP. Although the true incidence of these infections is unclear, given the limitations of the epidemiologic studies completed to date, viral NP represents a particular concern in immunocompromised patients. Future prospective multicenter studies are urgently needed to better delineate the epidemiology of this problem. Investigators have conducted such analyses in CAP and now should turn their attention to NP. Additionally, risk prediction models that focus on differing degrees of immunosuppression are needed to help facilitate both the rapid diagnosis of viral VAP as are clinical trials of novel therapeutics for these infections. If one does isolate a virus in NP while the bacterial cultures are negative, it may likely be safe to withhold antibiotics in this instance to promote antibiotic stewardship.

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
