# Peer review of "Respiratory Viruses in Nosocomial Pneumonia: An Evolving Paradigm"

_viruses, 2023, doi:10.3390/v15081676_

Round 1

Reviewer 1 Report

Zilberbeg et al. have provided a review on the potential relevance of viruses for nosocomial pneumonia. Considering the experience of the SARS-CoV-2 pandemic, the general fact of nosocomial virus transmission in the hospital setting is well known to clinicians. However, reviews summarizing available knowledge on the quantitative dimension of the phenomenon are rare.

Before publication may be considering, I have a few recommendations on how the work might be further improved.

1.) General comment: When speaking of community-acquired pneumonia (CAP), nosocomial pneumonia (NP), VAP (ventilator-associated pneumonia), HAP (hospital-acquired pneumonia), the authors should clearly provide their definitions of these terms and make it clear whether or not the authors of the quoted articles used these definitions accordingly.

2.) Each abbreviation should be spelled out at first use (for example, MV for mechanical ventilation).

3.) Line 38) The term “solely” seems a little bit hard. It has been known for decades to infection control specialists in hospitals that respiratory viruses like influence can nosocomially spread, making contact precautions necessary in the inpatient setting.

4.) Line 94: Why do the authors consider their finding “striking” that immunosuppressed patients are more vulnerable to respiratory virus infections? This is common sense.

5.) Line 105: Then evaluating the excess mortality, did the authors consider the rate of immunosuppression in the cohort? Was the prognosis better for immunosuppressed individuals with bacterial nosocomial pneumonia? Or did the mortality more reflect the generally poor health condition?

6.) Lines 119-122: The authors correctly mention testing-bias as a type of bias potentially leading to underestimation of viral contributions to nosocomial pneumonia. On the other hand, quality of testing for bacterial pathogens can be hampered if sampling is performed after the initiation of antibiotic therapy. Did some studies estimate the relevance of this problem? Might some cases of NP caused by “viruses only” be just due to missed bacterial pathogens under the condition of antibiotic treatment? I guess there is methodical bias from both directions.

7.) Line 186: Why not considering lower acquired immunity to respiratory viruses as a potential reason for a higher relevance of respiratory viruses in ill children?

8.) Lines 219-221: The authors feel that antibiotic drugs might be avoided if potential viral etiology of nosocomial pneumonia is considered. This is fine in theory, but which algorithm do they suggest considering the fact that viral etiology of nosocomial pneumonia is prevalent in less than 10% in the most studies? How to rapidly delineate the 10% viral nosocomial pneumonia from the 90% bacterial nosocomial pneumonia in the daily routine at a hospital to rapidly decide who will benefit from antibiotics and who will not?

9.) Conclusions: I feel that a reliable method of rapidly discriminating virally-induced from bacterially-induced nosocomial pneumonia is necessary if antibiotics shall be saved. If the authors have other ideas on this, maybe they want to explain them in the conclusions paragraph graph.

Author Response

We appreciate the opportunity to improve our manuscript by responding to Reviewer 1's comments.  Our responses are below.

  1. We appreciate the reviewer’s observation and have clarified this in the text.
  2. Each abbreviation is spelled out the first time it is used as a review of the text illustrates
  3. The word “solely” has been deleted as per the reviewer’s suggestion.
  4. We have deleted the word “striking.”
  5. We have clarified the issue of mortality as a function of viral vs bacterial pathogen. We have also clarified that this study (and the others) does not provide insight as to attributable vs crude mortality as no un-infected control group is reported for purposes of comparisons.
  6. We concur with the reviewer that bacterial cultures are imperfect. We now mention this in the text.
  7. The reviewer makes an excellent point regarding acquired immunity. We have revised the text to add this observation
  8. We have revised the sentence in question to address the reviewer’s comment.
  9. We have clarified our meaning in the conclusion. The point is that if you isolate a virus definitively and bacterial cultures are negative then it is likely safe to discontinue antibiotics.  We are not suggesting the withholding of antibiotics based on clinical criteria. 

Reviewer 2 Report

Overall, this is an interesting article on a highly relevant topic. However, I would recommend that the authors go through the text again and try to reorganize the information in a more structured manner. For example, it would be good to see the different numbers reported here presented as tables, to be able to compare across the age spectrum or across the syndrome spectrum, i.e., HAP vs. VAP, for example. Also, some further minor comments below:

-          Line 40: Please correct spelling for baumannii.

-          Line 41-42: Why “in vitro”?

-          Lines 91-92: The percentages quoted here are a bit unclear. Is this 57% out of the 32%?

-          Line 105: Is there a breakdown by viral vs bacterial mortality available to quote here?

-          Lines 126-127: “This number is strikingly consistent with the VAP study described earlier.” However, the authors state that this is from non-ICU patients, therefore, it is unlikely that this number reflects VAP, or does it?

-          Line 151: Please quote some estimates here and a literature reference.

-          Lines 156-158: It is very important that the vaccination status of the healthcare workers is discussed here. I recommend that the authors also explore the following references regarding the experience of HCW with COVID-19 vaccination (PMID: 34946473), and also the following reference regarding nosocomial spread of SARS-CoV-2 (PMID: 37409819)

-          Line 233: Several antivirals for influenza are already available.

Phrasing should be revised, for a better flow.

Author Response

We appreciate the opportunity to improve our manuscript by responding to Reviewer 2's comments.  Our responses are below.

  1. We appreciate the reviewer’s observation regarding a table. However, these studies are so heterogeneous as to make such a table inappropriate as a means for summarizing their findings.  This is especially true given that we approach the literature not only as looking at NP generally but also through exploring reports in specific populations (eg pediatrics) and of particular viruses.
  2. We have corrected the misspelling and appreciate the reviewer’s attention to detail.
  3. Studies of the impact of initial appropriate antibiotic therapy on mortality have always defined “appropriate” based on in vitro susceptibilities.
  4. We have clarified the sentence in question so now the meaning is more evident.
  5. We appreciate the confusion the reviewer notes – we have clarified the text.
  6. We have added text to clarify the difference in mortality in influenza for nosocomial vs community-onset cases.
  7. We appreciate the reviewer’s point and now discuss this issue and cite the references suggested.
  8. We have revised the text to reflect the reviewer’s point.